# Effects of Zinc Source and Level on the Intestinal Immunity of Xueshan Chickens under Heat Stress

**DOI:** 10.3390/ani13193025

**Published:** 2023-09-26

**Authors:** Jian Jin, Mengxiao Xue, Yuchen Tang, Liangliang Zhang, Ping Hu, Yun Hu, Demin Cai, Xugang Luo, Ming-an Sun

**Affiliations:** 1Institute of Comparative Medicine, College of Veterinary Medicine, Yangzhou University, Yangzhou 225009, China; dz120210008@yzu.edu.cn (J.J.); mx120231060@stu.yzu.edu.cn (Y.T.); mz120201482@stu.yzu.edu.cn (L.Z.); 2College of Animal Science and Technology, Yangzhou University, Yangzhou 225009, China; mz120211421@stu.yzu.edu.cn (M.X.); pinghu@yzu.edu.cn (P.H.); huyun@yzu.edu.cn (Y.H.); demincai@yzu.edu.cn (D.C.); 3Joint International Research Laboratory of Important Animal Infectious Diseases and Zoonoses of Jiangsu Higher Education Institutions, Yangzhou University, Yangzhou 225009, China; 4Jiangsu Co-Innovation Center for Prevention and Control of Important Animal Infectious Diseases and Zoonosis, Yangzhou University, Yangzhou 225009, China; 5Joint International Research Laboratory of Agriculture and Agri-Product Safety of Ministry of Education of China, Yangzhou University, Yangzhou 225009, China

**Keywords:** zinc, heat stress, Xueshan chicken, local breed, intestine, immunity

## Abstract

**Simple Summary:**

The intestinal tract is an important part of the immune system and forms a congenital barrier against food antigens and pathogenic microorganisms. In poultry, heat stress can induce intestinal mucosal injury, damage intestinal tight junctions and microvillous structures, and trigger an inflammatory response and enterogenic infection. As an essential trace element, zinc has been shown to mitigate the adverse effects of heat stress on broilers. However, how the dietary supplementation of different sources and levels of zinc can improve the heat stress capacity of Chinese landraces remains unclear. Here, we investigated the effects of different levels of zinc sulfate (ZnS) and zinc proteinate with moderate chelation strength (Zn-Prot M) on the intestinal immune function under heat stress in Xueshan chickens, which comprise an important local breed in China.

**Abstract:**

Heat stress can cause intestinal inflammation, impaired barrier integrity, and decreased immunity in poultry. While zinc is known to mitigate the adverse effects of heat stress, how the dietary supplementation of different sources and levels of it can improve the heat stress capacity of Chinese landraces remains unclear. This study investigated Xueshan chickens, which are an important local breed in China. The effects of different levels of ZnS and Zn-Prot M on their intestinal immune function under heat stress were compared. We found that different levels of ZnS and Zn-Prot M could effectively reduce the secretion level of IL-6 in the serum, and 60 mg/kg was optimal. Compared with ZnS, Zn-Prot M significantly increased duodenal villus height and VH/CD ratio, thus Zn-Prot M was more effective than ZnS. Both ZnS and Zn-Prot M significantly down-regulated *TNF-α*, *IL-1β*, and *MyD88* in 102-day-old duodenum, and *IL-1β*, *IL-6*, and *NFKBIA* in jejunum and ileum at 74, 88, and 102 days old, with 60 mg/kg Zn-Prot M determined as optimal. In conclusion, our study demonstrates that Zn-Prot M is superior to ZnS in improving intestinal immunity in Xueshan chickens, and 60 mg/kg is the optimal addition dose.

## 1. Introduction

Heat is a main challenge faced by the poultry industry, and heat stress is the most common stressor affecting poultry production standards [1]. Heat stress can cause physiological disorders and immune system damage and may even lead to death, thus causing huge economic losses to the poultry industry [2]. Broilers are more susceptible to heat stress because of their characteristics of high body temperature, sweat gland deficiency, and exuberant metabolism [3,4]. The intestine is not only an essential digestive organ but also a central part of the immune system that forms a congenital barrier against food antigens, pathogenic microorganisms, and other damages [5]. Heat stress can trigger intestinal mucosal damage, disrupt intestinal tight junctions and microvillous structures, and trigger inflammatory responses and enterogenic infections in poultry [6,7,8,9]. Under the existing broiler breeds and feeding and management conditions, compared with the non-nutritional strategies used to alleviate heat stress and reduce losses, the application of anti-heat stress additives is an effective nutritional management strategy to eliminate or alleviate heat stress, and commonly used anti-heat stress agents include trace elements, vitamins, electrolytes, and organic acids [10].

As a trace element involved in the production of metallothionein, zinc (Zn) plays an important role in a variety of biochemical processes. For instance, it is a catalytic cofactor of more than 300 enzymes and a structural component of hundreds of zinc finger protein transcription factors, playing a key role in enhancing immunity, growth, reproductive performance, and disease resistance [11,12]. Because there is insufficient Zn in natural feed ingredients, appropriate amounts of Zn are commonly added to poultry diets. The most commonly used Zn supplements in poultry feed are Zn sulfate (ZnS) and Zn oxide, which are of inorganic origin. In recent years, organic Zn sources for feed supplementation have become popular in livestock and poultry production [13]. The addition of Zn in poultry feed has a positive effect on the growth and immunity of broilers. Zn can enhance the physiological response, growth performance, and immune response of broilers, including promoting the proliferation and activity of immune cells (such as macrophages and lymphocytes), enhancing antioxidant capacity, and promoting the synthesis and secretion of antibodies [14,15]. In addition, Zn has been shown to positively modulate the intestinal microarchitecture in broilers raised at high ambient temperatures [16]. Adding Zn to the diet can improve the damage to the intestinal mucosal barrier function caused by Salmonella infection in broiler chickens [17]. Despite the fact that that Zn has been shown to reduce the adverse effects of heat stress on broilers, it is unclear whether the dietary addition of Zn, particularly Zn proteinate with moderate chelation strength (Zn-Prot M) and moderate chelating strength, can improve the heat stress capacity in the broilers of Chinese landraces.

Xueshan chickens are a new variety in China developed by crossbreeding high-quality local Tibetan chickens and Chahua chickens [18]. Their commercial importance is increasing due to their unique flavor and the texture of the meat. In this study, we investigated the effects of different sources and levels of Zn on intestinal immune function in Xueshan chickens under heat stress. We analyzed the secretion levels of immunoglobulin A (IgA), immunoglobulin G (IgG), immunoglobulin M (IgM), and interleukin-6 (IL-6) in serum via enzyme-linked immunosorbent assay, examined the effects of different zinc sources and levels on intestinal morphology via histopathology, and analyzed the mRNA expression levels of *IL-6*, *IL-1β*, *TNF-α*, *MyD88*, and *NFKBIA* genes in small intestinal tissues after zinc addition via a quantitative polymerase chain reaction. Overall, these results reveal the effects of different zinc sources and levels on intestinal immune function and protection in Xueshan chickens under heat stress.

## 2. Materials and Methods

### 2.1. Ethics Statement

The animal research proposal was approved by the Institutional Animal Care and Utilization Committee (IACUC) of the Animal Experimental Ethics Committee of Yangzhou University (Permit Number: SYXK (SU) IACUC 2012-0029).

### 2.2. Experimental Design

The experiment was conducted using 61-day-old Xueshan chickens, with the experimental period lasting for 42 days. A two-factor completely randomized trial design with 8 treatment groups was used: 1 (negative control group without Zn addition) + 1 (positive control group with ZnS addition to meet Zn requirements) + 2 (Zn source) × 3 (added Zn level) under heat stress conditions. The heat stress conditions for this test were set as 34 ± 1 °C, 9:00–17:00, 8 h/d, and the temperature for the remaining time periods was set as 28 ± 1 °C. The relative humidity under heat stress conditions was maintained at 55 ± 5% [19]. The basal diet without Zn supplementation group was used as the negative control group, and the 50 mg/kg inorganic ZnS group, which meets the Zn nutritional requirements of 61–102-day-old Xueshan chickens, was used as the positive control group. Three ZnS groups were fed diets supplemented with 30, 60, and 90 mg/kg of ZnS on the basis of the basal diet, and three Zn-Prot M groups were fed diets supplemented with 30, 60, and 90 mg/kg of Zn-Prot M on the basis of the basal diet, as detailed in Appendix A.

### 2.3. Experimental Diet Preparation

The nutritional requirements for broiler chicks recommended in the “Nutrient Requirements of Yellow Chickens (NY/T 3645–2020)” formulated by the National Technical Committee 274 on Animal Husbandry of Standardization Administration of China [20], corn–soybean meal basal diets without Zn supplementation (negative control diets), were prepared for 61–102-day-old broilers, and the diets for each treatment group were prepared by adding inorganic ZnS and Zn-Prot M, respectively, according to the experimental design and treatment settings in this diet. The two Zn sources were reagent-grade analytically pure inorganic ZnS (ZnSO_4_·7H_2_O, Zn content calculated as 22.62%) and Zn proteinate with moderate chelation strength (Zn-Prot M, 17.09% Zn, and Qf = 51.6 from analysis). In addition, to balance methionine and lysine levels in the diets of each treatment group, we also added synthetic methionine and lysine, as detailed in Appendix A.

### 2.4. Experimental Animals and Feeding Management

We purchased 512 healthy 50-day-old Xueshan chickens from Jiangsu Lihua Animal Husbandry Co., Ltd. During 50–60 days of age, all chickens were uniformly fed ad libitum with the same complete diet of corn–soybean meal (purchased from Jiangsu Lihua Animal Husbandry Co., Ltd., Changzhou, China). At 61 days of age, they were randomly divided into eight treatment groups, each with eight replicates of eight chickens. Chickens were housed in stainless steel cages, 90 × 70 × 45 cm in size. During the test, the animals were exposed to constant light for 24 h every day; tap water was provided ad libitum and drank; routine immunization was performed, and the welfare of the experimental animals was ensured. The health status of the chickens was observed and recorded daily during the experiment. If chickens became ill or died, a necropsy was performed immediately to determine the cause of death. The experimental area and chicken house were kept clean, sanitary, and disinfected; the chicken house was ventilated to maintain the temperature required for chicken growth; the feed was produced according to the standard without dampness, lumping or mold; and no other growth-promoting agents were added to the feed; the water was of good quality and pollution-free.

### 2.5. Sample Collection

At the ages of 74, 88, and 102 days during the experiment, after a 12 h overnight fasting (20:00 p.m. to 8:00 a.m. the next day) without restricting water intake, 5 mL of blood was drawn from the wing vein and hepatic portal vein of each chicken using a disposable blood collection needle. The blood was contained in non-anticoagulated vacuum tubes and centrifuged at 3000 r/min for 10 min. The serum was then collected and cryopreserved at −20 °C. The small intestines of the experimental chickens were removed; the duodenum samples were isolated; and the middle part of each intestinal segment was harvested to be larger than 2 cm and placed in 4% paraformaldehyde solution for a subsequent analysis of its morphology. The remaining intestine was washed with precooled normal saline, and the intestinal mucosa was scraped with a sterile glass slide and immediately placed in a sterilized 1.5 mL centrifuge tube, snap-frozen in liquid nitrogen, and stored in a −80 °C freezer.

### 2.6. ELISA Assay

The secretion levels of immunoglobulin IgA, IgG, IgM, and IL-6 in the serum of each sample were determined following the instructions of the ELISA kit (Beijing Solarbio Science & Technology Co., Ltd., Beijing, China).

### 2.7. RNA Extraction and cDNA Synthesis

Total RNA from tissues was extracted using Trizol reagent (Invitrogen, Carlsbad, CA, USA). The degree of RNA integrity was detected through 1% formaldehyde denaturing agarose gel electrophoresis. After the concentration and purity were determined using an ND-1000 nucleic acid/protein concentration tester, the RNA samples were stored at −80 °C until use. The reverse transcription experiment was performed with HiScript III All-in-one RT SuperMix Perfect for qPCR (Vazyme Biotech Co., Ltd., Nanjing, China).

### 2.8. qPCR

According to the gene sequence published in the GenBank database, qPCR primers were designed using Primer Premier 5.0 software. ACTB was used as a housekeeping gene for input normalization. Primer synthesis was conducted by Sangon Biotechnology (Shanghai, China), and the corresponding sequences are shown in Appendix A. qPCR analysis was performed with AceQ Universal SYBR qPCR Master Mix (Vazyme Biotech Co., Ltd., Nanjing, China). The results of relative quantification were analyzed and processed using the 2^−ΔΔCt^ method [21], and the expression levels were normalized to appropriate internal control genes.

### 2.9. Tissue Staining

The fixed intestinal tissues were subjected to standard paraffin embedding. Paraffin sections were prepared, and staining was performed using hematoxylin–eosin stain. The samples were observed and photographed using a Motic microscope (Motic, Wetzlar, Germany) and analyzed using the MShot Image Analysis System (Guangzhou, China). The system was used to measure villus height (VH), crypt depth (CD), and villus width (VW).

### 2.10. Statistical Analyses

Statistical analyses were performed via SAS statistical software (version 9.2, SAS Institute Inc., Cary, NC, USA). All data were analyzed using a single-degree-of-freedom contrast to compare all supplemental Zn treatments with the control. Data, excluding the control, were further analyzed as a 2 × 3 (Zn sources × Added Zn level) factorial arrangement of treatments by two-way ANOVA with a model including the main effects of Zn sources, Zn levels, and their interaction. Each replicate served as an experimental unit. If the variances were significant, the differences among the means were tested via the least significant difference method, and the statistical significance was set at *p* < 0.05.

## 3. Results

### 3.1. Effects of Zinc Sources and Levels on the Immunoglobulin and IL-6 Secretion in Serum

We first determined the effects of different zinc sources and supplementation levels on the immunoglobulin IgA, IgG, IgM, and IL-6 secretion levels in the serum of different-day-old Xueshan chickens under heat stress (Table 1). Our data demonstrate that at the age of 74 days, a dietary Zn supplementation with different Zn sources and levels significantly affected the secretion of IL-6, which was significantly higher in the 60 mg/kg group than the other groups, and significantly higher in the Zn-ProtM group than the ZnS group. With the increase in age, the dietary supplementation of different Zn sources and different levels of Zn and their interaction effects could effectively reduce the secretion level of IL-6 in the serum of Xueshan chickens at 102 days of age, and 60 mg/kg was the optimal amount. On the other hand, the supplement of zinc has minor effects on the levels of IgA, IgG, and IgM in serum.

### 3.2. Effects of Zinc Sources and Levels on the Intestinal Morphology

The effects of different Zn sources and Zn levels on the histomorphology of the small intestine of Xueshan chickens under heat stress were compared (Table 2). With the increase in age, the dietary supplementation of different Zn sources and different levels of Zn effectively decreased the crypt depth of the duodenum (*p* < 0.05) and increased the villus height and villus width. Compared with ZnS, the addition of Zn-Prot M significantly increased the duodenal villus height and V/C ratio. Therefore, Zn-Prot M remarkably outperforms ZnS in promoting the morphology of duodenum.

### 3.3. Effects of Zinc Sources and Levels on the Intestinal Expression of Immune-Related Genes

The expression levels of several inflammation-related genes (i.e., *TNF-α*, *IL-1β*, *IL-6*, *NFKBIA*, and *MyD88*) in the small intestine of Xueshan chickens were detected via qPCR (Table 3, Table 4 and Table 5). The results demonstrate that the Zn source significantly up-regulated the mRNA expression levels of *IL-6* in the duodenum of the Xueshan chickens aged 74 and 88 days (*p* < 0.05), but it had no significant effect on the mRNA expression levels of *TNF-α*, *IL-1β*, *NFKBIA*, and *MyD88*. However, the Zn source significantly down-regulated the mRNA expression of *TNF-α*, *IL-1β*, and *MyD88* in the duodenum of the 102-day-old Xueshan chickens (*p* < 0.05). In jejunum and ileum, the different Zn sources and different levels of Zn could down-regulate the expression of the *IL-1β*, *IL-6*, and *NFKBIA* genes at 74, 88 and, 102 days old, and the addition of 60 mg/kg Zn-Prot M showed an optimal effect.

## 4. Discussion

The role that the trace element Zn plays in animal health has been demonstrated, but this substance cannot be stored in the body and therefore requires a regular intake of Zn to meet the functional requirements [22,23]. The small intestine is an important organ for absorbing nutrients, and the height of the small intestinal villi and crypt depth are important indicators to measure the absorption capacity of the small intestine. The villus height was positively correlated with the number of epithelial cells. When the villus height increased, the number of epithelial cells and the ability to absorb nutrients increased. Crypt depth can reflect the maturity and proliferation rate of epithelial cells and crypt shallowing, as well as enhance cell maturity and absorption nutrient capacity. The ratio of VH/CD can comprehensively reflect the absorption capacity of the small intestine; the absorption capacity ratio is enhanced [24,25]. It has been shown that heat stress adversely impacts intestinal health by modulating intestinal morphology, increasing crypt depth, and decreasing villus height and VH/CD ratio [26]. Zn supplementation can enhance intestinal immune function, improve intestinal mucosal barrier function, and improve anti-inflammatory capacity [27,28], which is important for intestinal health and immune protection in animals under heat stress. Tako et al. showed that Zn can promote cell division, proliferation, and protein synthesis and also improve small intestinal morphology [29]. Ma et al. showed that feeding broilers a diet containing 90 mg/kg Zn glycine significantly increased duodenal villus height and decreased crypt depth [30]. In this study, Xueshan chicken, a specific local chicken breed, was selected as the experimental object to investigate the effect of different levels of Zn in the diet on the intestinal tract of these chickens under heat stress. Our findings showed villus growth and crypt shallowing following Zn addition under heat stress, suggesting that Zn may contribute to attenuating heat-stress-induced intestinal injury. Compared with ZnS, Zn-Prot M could increase the height of small intestinal villi and enhance the ratio of VH/CD, indicating that Zn-Prot M was slightly more effective than ZnS in its effect on small intestinal morphology, but the difference between Zn-Prot M and ZnS was not significant in the test, which may be because the addition level of Zn was the same in each group, while the height crypt depth of small intestinal villi in broilers during the test period was relatively short in response to the difference between Zn sources, so the improvement effect on small intestinal morphology could not be fully reflected.

Under heat stress conditions, the expression levels of *IL-1β*, *IL-6*, *TNF-α*, *NFKBIA*, and *MyD88* in the intestine of Xueshan chickens can be assessed through qPCR to provide an evaluation of intestinal immune characteristics [31,32,33,34,35]. These genes are important molecules closely related to immune response and inflammation regulation. IL-1β and IL-6 are pro-inflammatory cytokines that play a role in the regulation and transmission of inflammatory responses and various cellular activities. An overexpression of IL-1β and IL-6 is closely associated with the occurrence and development of inflammatory diseases [36]. Therefore, the observation of the decreased expression levels of IL-1β and IL-6 after organic Zn supplementation may suggest that organic Zn may have an inhibitory effect on the inflammatory response. Organic Zn may regulate the relevant signaling pathways to inhibit the production of IL-1β and IL-6 or decrease their expression levels, thus alleviating the inflammatory response and immune abnormalities. NFKBIA is an inhibitory protein of nuclear factor-κB (NF-κB), which plays an important role in regulating the inflammatory signaling pathway. NF-κB is a key transcription factor involved in the regulation of inflammatory gene expression [37], and a decrease in NFKBIA expression level is often associated with an activation of the NF-κB signaling pathway. Therefore, the observation of the decreased expression level of NFKBIA after organic Zn supplementation may indicate that organic Zn may regulate the inflammatory response by inhibiting the activation of the NF-κB signaling pathway. These experimental results suggest that adding organic Zn under heat stress conditions can reduce the expression levels of *IL-1β*, *IL-6*, and *NFKBIA*, thereby alleviating the inflammatory response and immune abnormalities. This helps maintain the health of the intestine, protect the integrity of the intestinal mucosal barrier, and improve the animal’s heat stress adaptability.

While this study focused on Xueshan chickens, which are a Chinese breed, further research on the mechanism of Zn in intestinal immune regulation has valuable implications for other chicken breeds as well. Different chicken breeds may have differences in their physiological characteristics and immune requirements, so further studies are needed to develop nutritional strategies tailored to their specific needs. Additionally, this study only observed changes in the expression levels of immune genes, and the specific impacts and regulatory mechanisms of Zn on intestinal immune mechanisms are still unclear. Further research is required to explore these mechanisms.

## 5. Conclusions

In summary, this study investigated the effects of different zinc sources and levels on the intestinal immunity of Xueshan chickens under heat stress. Our results indicate that adding specific sources and levels of zinc to the feed can improve their intestinal immune function under heat stress conditions. Specifically, Zn-Prot M is more effective than ZnS, with an optimal supplementation level of 60 mg/kg.

## Figures and Tables

**Table 1 animals-13-03025-t001:** Effects of zinc source and added zinc level on immunoglobulin and IL-6 secretion levels in 74-, 88-, and 102-day-old Xueshan chickens under heat stress.

Items	Zn Level (mg/kg)	74 Days Old				88 Days Old				102 Days Old			
		IgA (mg/mL)	IgG (mg/mL)	IgM (mg/mL)	IL-6 (pg/mL)	IgA (mg/mL)	IgG (mg/mL)	IgM (mg/mL)	IL-6 (pg/mL)	IgA (mg/mL)	IgG (mg/mL)	IgM (mg/mL)	IL-6 (pg/mL)
Control	0	0.13	17.97	0.17	7.29	0.14	19.84	0.12	6.48	0.55	19.57	0.14	12.06
ZnS Control	50	0.19	21.74	0.18	11.00	0.16	20.61	0.27	12.03	0.36	19.10	0.17	8.37
ZnS	30	0.17	19.93 ^ab^	0.16	10.66 ^b^	0.21	22.24	0.23 ^a^	27.25 ^a^	0.42	18.09	0.19 ^b^	4.78 ^b^
	60	0.19	20.31 ^ab^	0.19	15.04 ^b^	0.20	19.68	0.23 ^a^	14.83 ^b^	0.43	15.55	0.18 ^b^	5.53 ^b^
	90	0.21	24.36 ^a^	0.18	6.90 ^b^	0.16	19.52	0.13 ^b^	4.04 ^c^	0.28	15.40	0.37 ^a^	6.36 ^b^
Zn-Prot M	30	0.18	21.15 ^ab^	0.19	38.30 ^b^	0.20	20.55	0.15 ^b^	5.30 ^bc^	0.70	17.49	0.31 ^a^	21.19 ^a^
	60	0.20	24.26 ^a^	0.17	149.12 ^a^	0.19	17.81	0.15 ^b^	5.81 ^bc^	0.42	18.05	0.16 ^b^	4.91 ^b^
	90	0.15	15.52 ^b^	0.16	151.03 ^a^	0.20	18.92	0.17 ^ab^	14.81 ^b^	0.41	20.33	0.14 ^b^	5.65 ^b^
SEM		0.0232	3.5385	0.0217	12.2713	0.0211	1.7642	0.0211	2.9464	0.0527	1.5387	0.0237	1.3608
Zn source	ZnS	0.19	21.41	0.17	10.86 ^b^	0.19	20.51	0.20 ^a^	14.47 ^a^	0.38 ^b^	16.35	0.22	5.56 ^b^
	Zn-Prot M	0.18	20.52	0.17	112.82 ^a^	0.20	19.03	0.16 ^b^	8.04 ^b^	0.50 ^a^	18.62	0.19	10.09 ^a^
SEM		0.0150	1.2984	0.0135	7.0848	0.0125	1.0425	0.0128	1.9438	0.0319	0.8883	0.0148	0.8958
Zn level	30	0.18	20.45	0.17	24.48 ^b^	0.20	21.45	0.20	15.06	0.53 ^a^	17.79	0.24 ^a^	12.07 ^a^
	60	0.19	22.15	0.18	82.08 ^a^	0.20	18.68	0.19	10.32	0.42 ^b^	16.80	0.17 ^b^	5.15 ^b^
	90	0.18	20.28	0.17	78.96 ^a^	0.18	19.22	0.15	8.08	0.34 ^b^	17.87	0.20 ^a^	6.09 ^b^
SEM		0.0187	1.6082	0.0171	8.6771	0.0154	1.2475	0.0161	2.2099	0.0415	1.0880	0.0180	1.0758
*p*-value	Zn source	0.5532	0.5312	0.9823	<0.0001	0.6284	0.3522	0.0357	0.0290	0.0121	0.0773	0.0584	0.0010
	Zn level	0.8581	0.5857	0.9417	<0.0001	0.5572	0.3115	0.1852	0.1035	0.0033	0.7424	0.0030	0.0001
	Zn source × Zn level	0.4128	0.0276	0.5482	0.0001	0.3453	0.9285	0.0147	0.0004	0.0824	0.2101	< 0.0001	< 0.0001

^a–c^ Means with different superscripts within the same column differ significantly (*p* < 0.05).

**Table 2 animals-13-03025-t002:** Effect of zinc source and added zinc level on duodenum morphology of 74-, 88-, and 102-day-old Xueshan chickens under heat stress.

Items	Zn Level (mg/kg)	74 Days Old				88 Days Old				102 Days Old			
		CD (μm)	VH (μm)	VW (μm)	VH/CD	CD (μm)	VH (μm)	VW (μm)	VH/CD	CD (μm)	VH (μm)	VW (μm)	VH/CD
Control	0	237.84	1346.75	337.54	5.74	221.22	1240.46	329.35	5.93	237.16	1165.14	310.00	5.14
ZnS Control	50	284.49	1413.00	398.50	4.99	324.63	1285.13	326.98	3.99	258.47	1213.27	304.14	4.73
ZnS	30	224.29	1389.56	339.06	6.66	244.20	1323.86	328.99	5.78	243.04 ^a^	1251.15 ^c^	349.19 ^b^	5.25 ^bc^
	60	244.43	1376.95	318.91	5.70	265.55	1251.00	346.02	4.77	255.90 ^ab^	1159.23 ^c^	295.73 ^c^	4.60 ^c^
	90	259.92	1557.42	373.38	6.12	283.11	1292.10	319.57	4.72	209.34 ^bc^	1245.41 ^c^	278.17 ^c^	6.00 ^b^
Zn-Prot M	30	245.46	1314.00	330.31	5.60	216.23	1319.14	327.02	6.20	185.91 ^c^	1398.41 ^b^	400.23 ^a^	7.58 ^a^
	60	257.39	1324.52	315.80	5.18	208.55	1307.59	381.28	6.38	205.53 ^c^	1578.57 ^a^	264.86 ^c^	7.71 ^a^
	90	266.00	1409.16	340.59	5.52	283.92	1320.66	351.27	4.68	294.81 ^a^	1495.62 ^b^	255.48 ^c^	5.34 ^bc^
SEM		18.86	29.27	11.01	0.52	16.21	47.00	17.38	0.44	17.27	44.57	17.49	0.36
Zn source	ZnS	242.88	1441.31 ^a^	343.79	6.16	264.29 ^a^	1288.98	331.53	5.13	236.09	1218.60 ^b^	307.70	5.28 ^b^
	Zn-Prot M	256.29	1349.23 ^b^	328.90	5.43	236.23 ^b^	1315.80	353.19	5.59	228.75	1490.87 ^a^	306.86	6.88 ^a^
SEM		10.89	16.90	6.36	0.30	9.36	27.13	10.04	0.26	9.97	25.73	10.10	0.21
Zn level	30	234.88	1351.78 ^b^	334.68 ^ab^	6.13	230.22 ^b^	1321.50	328.01	5.91 ^a^	214.47	1324.78	374.71 ^a^	6.41
	60	250.91	1350.73 ^b^	317.36 ^b^	5.44	237.05 ^b^	1279.29	363.65	5.51 ^ab^	230.71	1368.90	280.29 ^b^	6.16
	90	262.96	1483.29 ^a^	356.99 ^a^	5.82	283.51 ^a^	1306.38	335.42	4.67 ^b^	252.08	1370.51	266.83 ^b^	5.67
SEM		13.34	20.70	7.79	0.37	11.46	33.23	12.29	0.31	12.21	31.52	12.37	0.25
*p*-value	Zn source	0.3910	0.0006	0.1082	0.0954	0.0424	0.4901	0.1373	0.2057	0.6065	<0.0001	0.9533	<0.0001
	Zn level	0.3408	<0.0001	0.0045	0.4232	0.0048	0.6648	0.1136	0.0263	0.1094	0.5153	<0.0001	0.1288
	Zn source × Zn level	0.9231	0.2485	0.3713	0.8552	0.2205	0.8092	0.5052	0.1918	0.0003	0.0161	0.0496	<0.0001

^a–c^ Means with different superscripts within the same column differ significantly (*p* < 0.05). CD, crypt depth; VH, villus heigh; VW, villus width.

**Table 3 animals-13-03025-t003:** Effect of zinc source and added zinc level on the expression level of immune-related genes in the duodenum of 74-, 88-, and 102-day-old Xueshan chickens under heat stress.

Items	Zn Level (mg/kg)	74 Days Old					88 Days Old					102 Days Old				
		TNF-α	IL-1β	IL-6	NFKBIA	MyD88	TNF-α	IL-1β	IL-6	NFKBIA	MyD88	TNF-α	IL-1β	IL-6	NFKBIA	MyD88
Control	0	1.02	0.61	0.98	1.07	1.04	0.82	0.73	0.77	1.16	1.18	0.77	1.89	42.81	1.47	1.77
ZnS Control	50	1.06	1.27	0.49	1.17	1.50	0.85	1.19	0.84	0.92	1.22	0.83	2.97	21.67	1.27	1.28
ZnS	30	1.22	0.40	0.49 ^e^	0.67	1.05	0.96 ^a^	0.88 ^b^	0.74 ^c^	1.00	0.89 ^b^	0.75	6.63 ^a^	16.98	1.59 ^a^	2.33 ^a^
	60	0.88	0.94	0.63 ^de^	0.86	1.00	0.85 ^ab^	0.61 ^b^	0.91 ^bc^	0.99	0.90 ^b^	0.75	2.62 ^bc^	19.87	1.30 ^ab^	1.19 ^b^
	90	1.25	2.26	2.21 ^cd^	1.19	1.33	0.63 ^bc^	2.65 ^a^	7.28 ^a^	1.34	1.44 ^a^	0.80	3.89 ^b^	26.53	1.21 ^ab^	1.41 ^b^
Zn-Prot M	30	1.00	1.12	8.46 ^a^	1.02	0.97	0.91 ^a^	2.63 ^a^	1.26 ^b^	1.11	1.23 ^a^	0.57	1.36 ^c^	20.29	0.91 ^b^	1.00 ^b^
	60	0.89	1.00	3.14 ^bc^	0.93	1.00	0.58 ^c^	0.68 ^b^	1.26 ^b^	0.96	0.78 ^b^	0.62	1.90 ^bc^	27.20	1.43 ^a^	1.40 ^b^
	90	0.96	1.97	4.52 ^b^	0.95	1.49	0.82 ^ab^	0.55 ^b^	0.89 ^bc^	1.00	0.80 ^b^	0.57	2.69 ^bc^	13.52	1.52 ^a^	1.40 ^b^
SEM		0.12	0.16	0.54	0.11	0.12	0.0800	0.1329	0.1592	0.1084	0.1131	0.0908	0.6580	7.4750	0.1302	0.1510
Zn source	ZnS	1.12	1.22	1.11 ^b^	0.92	1.11	0.81	1.26	2.44 ^a^	1.11	1.08	0.77 ^a^	4.01 ^a^	20.49	1.36	1.55 ^a^
	Zn-Prot M	0.95	1.29	5.45 ^a^	0.96	1.16	0.77	1.07	1.14 ^b^	1.01	0.93	0.58 ^b^	1.90 ^b^	19.20	1.28	1.26 ^b^
SEM		0.07	0.11	0.31	0.07	0.07	0.0462	0.0809	0.1002	0.0626	0.0653	0.0524	0.4340	5.1064	0.0792	0.0892
Zn level	30	1.12	0.72 ^b^	4.47 ^a^	0.86	1.01 ^b^	0.94 ^a^	1.46 ^a^	0.98 ^b^	1.04	1.06 ^ab^	0.65	3.47	18.48	1.25	1.51
	60	0.89	0.97 ^b^	1.88 ^b^	0.90	1.00 ^b^	0.72 ^b^	0.65 ^b^	1.10 ^b^	0.98	0.84 ^b^	0.68	2.36	23.53	1.36	1.29
	90	1.10	2.14 ^a^	3.20 ^a^	1.06	1.42^a^	0.73 ^b^	1.60 ^a^	3.45 ^a^	1.17	1.12 ^a^	0.69	3.41	17.86	1.35	1.40
SEM		0.08	0.13	0.38	0.08	0.08	0.0566	0.0940	0.1172	0.0766	0.0800	0.0642	0.5455	6.4012	0.0921	0.1068
*p*-value	Zn source	0.0928	0.3235	<0.0001	0.5320	0.7663	0.5255	0.4493	<0.0001	0.3436	0.1337	0.0201	0.0010	0.9169	0.4906	0.0068
	Zn level	0.1163	<0.0001	0.0007	0.1526	0.0026	0.0132	<0.0001	<0.0001	0.2147	0.0439	0.9516	0.1023	0.8638	0.6519	0.0768
	Zn source × Zn level	0.4188	0.0576	<0.0001	0.0596	0.6163	0.0230	<0.0001	<0.0001	0.1294	0.0004	0.8517	0.0147	0.5146	0.0032	<0.0001

^a–e^ Means with different superscripts within the same column differ significantly (*p* < 0.05).

**Table 4 animals-13-03025-t004:** Effect of zinc source and added zinc level on the expression level of immune-related genes in the jejunum of 74, 88-, and 102-day-old Xueshan chickens under heat stress.

Items	Zn Level (mg/kg)	74 Days Old					88 Days Old					102 Days Old				
		TNF-α	IL-1β	IL-6	NFKBIA	MyD88	TNF-α	IL-1β	IL-6	NFKBIA	MyD88	TNF-α	IL-1β	IL-6	NFKBIA	MyD88
Control	0	1.01	0.40	0.87	1.08	1.07	0.75	40.06	13.30	0.77	0.72	1.01	28.31	29.62	0.99	0.80
ZnS Control	50	1.13	2.12	0.86	0.85	0.99	0.97	0.96	12.41	1.09	0.65	0.74	6.69	12.96	0.62	1.97
ZnS	30	0.83	0.87 ^b^	1.11 ^bc^	0.99	0.98	0.62 ^c^	1.14	6.03 ^a^	0.84	0.86	0.62	332.72 ^a^	236.02	0.89	2.75 ^a^
	60	0.86	0.92 ^b^	1.54 ^bc^	0.73	0.60	0.69 ^c^	0.61	2.56 ^b^	0.78	0.69	0.87	30.16 ^b^	2.21	1.00	0.69 ^d^
	90	0.68	1.59 ^a^	8.86 ^a^	0.78	0.70	0.70 ^bc^	0.65	0.95 ^c^	0.69	0.56	0.84	43.76 ^b^	36.37	0.86	0.96 ^cd^
Zn-Prot M	30	0.76	1.02 ^ab^	3.60 ^bc^	0.81	0.61	1.09 ^a^	0.51	0.89 ^c^	0.83	0.63	0.80	54.38 ^b^	169.64	0.78	0.65 ^d^
	60	0.95	0.62 ^b^	4.19 ^b^	0.80	0.75	0.73 ^bc^	0.38	0.66 ^c^	0.62	0.45	0.85	0.90 ^b^	47.35	0.76	2.09 ^b^
	90	0.79	0.78 ^b^	0.83 ^c^	0.70	0.64	0.91 ^ab^	0.32	0.52 ^c^	0.58	0.50	0.98	38.19 ^b^	5.45	0.99	1.29 ^c^
SEM		0.0923	0.1523	0.9707	0.1205	0.1058	0.0746	0.1066	0.3173	0.0663	0.0714	0.1121	24.3820	42.8359	0.1337	0.1761
Zn source	ZnS	0.79	1.11	3.63	0.82	0.74	0.67 ^b^	0.79 ^a^	3.18 ^a^	0.77	0.71 ^a^	0.77	140.82 ^a^	91.53	0.91	1.33
	Zn-Prot M	0.83	0.82	2.46	0.77	0.67	0.91 ^a^	0.39 ^b^	0.66 ^b^	0.68	0.52 ^b^	0.87	33.32 ^b^	80.11	0.81	1.20
SEM		0.0533	0.1057	0.6864	0.0713	0.0626	0.0430	0.0644	0.2046	0.0383	0.0410	0.0663	14.4627	26.3285	0.0808	0.1082
Zn level	30	0.79	0.95	2.22	0.90	0.78	0.85	0.85 ^a^	3.46 ^a^	0.84 ^a^	0.74 ^a^	0.71	216.74 ^a^	199.81 ^a^	0.83	1.52 ^a^
	60	0.91	0.77	2.87	0.76	0.67	0.71	0.49 ^b^	1.45 ^b^	0.70 ^b^	0.57 ^b^	0.86	18.45 ^b^	24.78 ^b^	0.87	1.20 ^ab^
	90	0.74	1.05	3.75	0.74	0.67	0.81	0.49 ^b^	0.69 ^c^	0.63 ^b^	0.53 ^b^	0.91	41.44 ^b^	20.91 ^b^	0.91	1.08 ^b^
SEM		0.0653	0.1305	0.8049	0.0853	0.0749	0.0527	0.0754	0.2558	0.0469	0.0502	0.0793	18.8862	31.7679	0.0979	0.1364
*p*-value	Zn source	0.5995	0.0568	0.3397	0.5466	0.2989	0.0003	<0.0001	<0.0001	0.0984	0.0033	0.2822	<0.0001	0.6493	0.5397	0.4521
	Zn level	0.1854	0.1292	0.1042	0.4080	0.4655	0.1637	0.0079	<0.0001	0.0114	0.0115	0.2112	<0.0001	0.0005	0.8242	0.0252
	Zn source × Zn level	0.5952	0.0487	0.0001	0.6177	0.0827	0.0197	0.2126	<0.0001	0.5024	0.3609	0.6321	<0.0001	0.4776	0.4786	<0.0001

^a–d^ Means with different superscripts within the same column differ significantly (*p* < 0.05).

**Table 5 animals-13-03025-t005:** Effect of zinc source and added zinc level on the expression level of immune-related genes in the ileum of 74-, 88-, and 102-day-old Xueshan chickens under heat stress.

Items	Zn Level (mg/kg)	74 Days Old					88 Days Old					102 Days Old				
		TNF-α	IL-1β	IL-6	NFKBIA	MyD88	TNF-α	IL-1β	IL-6	NFKBIA	MyD88	TNF-α	IL-1β	IL-6	NFKBIA	MyD88
Control	0	1.01	1.10	0.73	1.09	1.09	1.12	1.14	0.78	1.14	0.84	0.69	1.96	2.26	1.35	1.05
ZnS Control	50	1.18	0.81	0.92	0.95	0.90	1.11	1.14	1.35	1.12	0.98	0.88	3.70	3.31	1.55	0.92
ZnS	30	1.04	1.07	0.96 ^bc^	0.95	0.84	0.90	0.88	0.80 ^b^	1.03 ^b^	0.57	0.65 ^bc^	5.76 ^b^	3.83 ^b^	1.66 ^b^	0.92
	60	1.10	1.00	0.44 ^c^	0.84	0.68	0.89	0.92	0.96 ^ab^	0.80 ^b^	0.65	0.82 ^ab^	5.82 ^b^	0.85 ^b^	2.50 ^a^	0.91
	90	1.02	0.80	0.67 ^b^	0.96	0.74	0.87	0.73	0.40 ^c^	0.98 ^b^	0.44	1.02 ^a^	155.83 ^a^	18.36 ^a^	1.20 ^b^	0.68
Zn-Prot M	30	0.95	1.57	0.84 ^bc^	0.94	0.74	0.95	1.14	0.30 ^c^	22.48 ^a^	0.62	0.97 ^a^	1.94 ^b^	0.58 ^b^	1.24 ^b^	1.01
	60	1.28	0.69	1.71 ^a^	0.94	0.85	0.94	0.67	0.26 ^c^	0.97 ^b^	0.65	1.02 ^a^	1.98 ^b^	0.77 ^b^	1.45 ^b^	1.14
	90	1.09	0.61	1.19 ^ab^	0.82	0.89	0.79	1.07	1.31 ^a^	0.98 ^b^	0.53	0.67 ^b^	3.37 ^b^	0.47 ^b^	1.30 ^b^	0.97
SEM		0.1273	0.1762	0.1824	0.1080	0.0774	0.0947	0.1401	0.1097	0.3322	0.0772	0.0957	14.7957	1.0070	0.1629	0.1326
Zn source	ZnS	1.05	0.96	0.65 ^b^	0.91	0.75	0.89	0.84	0.68	0.94 ^b^	0.55	0.82	41.10 ^a^	5.85 ^a^	1.69 ^a^	0.85
	Zn-Prot M	1.11	0.88	1.25 ^a^	0.90	0.83	0.89	0.94	0.62	5.50 ^a^	0.60	0.89	2.33 ^b^	0.60 ^b^	1.35 ^b^	1.04
SEM		0.0735	0.1134	0.1154	0.0638	0.0457	0.0547	0.0828	0.0761	0.2007	0.0446	0.0552	10.5777	0.7147	0.1098	0.0807
Zn level	30	1.00	1.32 ^a^	0.91	0.94	0.79	0.92	0.97	0.55	8.18 ^a^	0.59	0.81	3.67 ^b^	2.20 ^b^	1.47 ^b^	0.96
	60	1.19	0.85 ^b^	0.86	0.89	0.76	0.91	0.80	0.56	0.89 ^b^	0.65	0.92	4.34 ^b^	0.82 ^b^	1.83 ^a^	1.01
	90	1.06	0.70 ^b^	0.84	0.89	0.82	0.83	0.90	0.79	0.98 ^b^	0.48	0.84	79.60 ^a^	7.63 ^a^	1.24 ^b^	0.83
SEM		0.0900	0.1244	0.1580	0.0764	0.0547	0.0670	0.0990	0.0838	0.3696	0.0546	0.0677	11.9287	0.8119	0.1306	0.0938
*p*-value	Zn source	0.6031	0.9988	0.0063	0.8536	0.2696	0.9505	0.3642	0.3671	<0.0001	0.4544	0.4636	0.0017	<0.0001	0.0073	0.0883
	Zn level	0.3077	0.0184	0.7186	0.8764	0.8018	0.5721	0.4255	0.0536	<0.0001	0.1177	0.5200	0.0007	<0.0001	0.0026	0.3989
	Zn source × Zn level	0.5895	0.1285	0.0210	0.5318	0.2119	0.7253	0.1191	<0.0001	<0.0001	0.8051	0.0028	0.0009	<0.0001	0.0222	0.7498

^a–c^ Means with different superscripts within the same column differ significantly (*p* < 0.05).

## Data Availability

Not applicable.

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
