# Peer review of "Effects of Zinc Source and Level on the Intestinal Immunity of Xueshan Chickens under Heat Stress"

_animals, 2023, doi:10.3390/ani13193025_

Round 1

Reviewer 1 Report

1. In the abstract, materials and methods was missing. Please provide information regarding experimental design and methodologies.

2. line 68-69, “Currently, Zn in poultry…….. zinc sulfate (ZnS)”. I think this statement is not appropriate, as we all know that zinc oxide is also widely used in poultry. Please change it.

3. Please define the abbreviation correctly, for example, line 81 “Zinc”→ “Zn”; line 82, the first appearance of IgA, IgG, IgM and IL-6 should define them.

4. In the experimental design, the addition amount of zinc from two different sources is 30,60 and 90mg/kg. How to ensure the consistent Zn content in the same level of zinc sources? What is the reason for choosing 61-day-old Xueshan chickens for the experiment?

5. It is recommended to give growth performance data.

6. Gens should be italics, for example, line 223 TNF-α, IL-1β, IL-6, NFKBIA and MyD88.

Minor editing of English language required.

Author Response

Response to reviewer’s comments:

  1. In the abstract, materials and methods was missing. Please provide information regarding experimental design and methodologies.

Response:

Thanks for your comments. We have made the appropriate modifications to the abstract.

  1. line 68-69, “Currently, Zn in poultry…….. zinc sulfate (ZnS)”. I think this statement is not appropriate, as we all know that zinc oxide is also widely used in poultry. Please change it.

Response:

Thanks for your advice. We have extensively re-wrote that paragraph as you suggested, and we believe it is clearer at current form now. The updated paragraph is as below:

The most commonly used Zn supplements in poultry feed are Zn sulfate (ZnS) and Zn oxide which are inorganic origin. In recent years, organic Zn sources for feed supplementation have become popular in livestock and poultry production”.

  1. Please define the abbreviation correctly, for example, line 81 “Zinc”→ “Zn”; line 82, the first appearance of IgA, IgG, IgM and IL-6 should define them.

Response:

We have made corresponding modifications in the manuscript.

  1. In the experimental design, the addition amount of zinc from two different sources is 30,60 and 90mg/kg. How to ensure the consistent Zn content in the same level of zinc sources? What is the reason for choosing 61-day-old Xueshan chickens for the experiment?

Response:

Thanks for your insightful comment. We chose high-quality zinc supplements to ensure stable quality and satisfactory zinc content from two different sources. Each zinc supplement was proportioned and calibrated according to the required amount added to ensure accurate proportioning to achieve the desired zinc content. For each treatment group, we added zinc supplements in strict accordance with the formulated formulations and procedures to ensure error minimization. In each treatment group, experimental samples were analyzed for zinc content to ensure that zinc content met the expected requirements.

At around 61 days of age, Xueshan chickens are entering the late growth phase, and the adaptability and immune system development of Xueshan chickens have been relatively perfect during this period. At the same time, the sensitivity and stress response of Xueshan chickens under heat stress conditions were also more obvious. Therefore, selecting this specific period for testing could better assess the effect of zinc supplementation on Xueshan chickens under heat stress.

  1. It is recommended to give growth performance data.

Response:

Thanks for your advice. We agree that growth performance is important. However, this study focused on the intestinal immunity after heat stress, and the growth performance data are not fully documented since. We and collaborators are performing additional studies on the effect of zinc to Xueshan chickens, and we will pay more attention about the growth performance.

  1. Genes should be italics, for example, line 223 TNF-α, IL-1β, IL-6, NFKBIA and MyD88.

Response:

We have made corresponding modifications in the manuscript.

Reviewer 2 Report

Good afternoon!

Many thanks for submitting an interesting and scientific manuscript to the Journal. The topic of zinc research has recently become relevant (different forms under different microclimate conditions). In general, evaluating the manuscript positively, I would like to note a few comments:

1. In Table 1, could you put it in a simpler form. Because it is not entirely clear which data belong to which group (because there are no horizontal lines).

2. What is the reason for the choice of these dosages of zinc?

3. It was necessary to give the composition of compound feed and its nutritional value. In particular, the content of mineral substances. Because we cannot assess whether the diets were balanced or not?What was the zinc level in the groups?

4. In Table 2, it is similar in Table 1. Some data have moved out, it is not entirely clear what they relate to?

5. Have you taken into account the level of zinc in the feed?

6. Regarding the discussion, the mechanism of action of your supplements and how they work is not explained.

7. Conclusions need to be drawn so that they are understandable to the public.

8. What is the reason that you have chosen this breed as the object of study of chickens.

9. It is advisable to also indicate the productivity of poultry in the article, because in my opinion, all studies are indirectly or directly aimed at improving or increasing productivity.

10. The title of the article should be done differently for you more narrowly.

11. The list of references should be drawn up according to the requirements of the journal.

Author Response

Response to reviewer’s comments:

  1. In Table 1, could you put it in a simpler form. Because it is not entirely clear which data belong to which group (because there are no horizontal lines).

Response:

Thanks for your advice. We have reformatted Table 1 to make the data clearer.

  1. What is the reason for the choice of these dosages of zinc?

Response:

In general, the zinc requirement of broilers was 30-50 mg/kg, and the dietary zinc requirement of Chinese yellow-feathered laying hens was 72 mg/kg [1]. In this study, the experimental zinc concentrations of 30, 60 and 90 mg/kg were designed according to the relevant literature reports and in combination with the actual situation to determine the optimal dietary zinc requirement of Xueshan chickens.

  1. Li, L.; Abouelezz, K.F.M.; Gou, Z.; Lin, X.; Wang, Y.; Fan, Q.; Cheng, Z.; Ding, F.; Jiang, S.; Jiang, Z. Optimization of Dietary Zinc Requirement for Broiler Breeder Hens of Chinese Yellow-Feathered Chicken. Animals (Basel). 2019, 9, 472.

  1. It was necessary to give the composition of compound feed and its nutritional value. In particular, the content of mineral substances. Because we cannot assess whether the diets were balanced or not? What was the zinc level in the groups?

Response:

Thanks for your suggestion. We have provided the composition of the formulated feed and its nutritional value in the supplementary Table 2.

The basal diet without Zn supplementation was used as the negative control group, and 50 mg/kg inorganic ZnS to meet the Zn nutritional requirements of 61–102-day-old Xueshan chickens was used as the positive control group. Three ZnS groups were fed diets supplemented with 30, 60, and 90 mg/kg ZnS on the basis of the basal diet, and three Zn-Prot M groups were fed diets supplemented with 30, 60, and 90 mg/kg Zn-Prot M on the basis of the basal diet, as detailed in Supplementary Table 1.

  1. In Table 2, it is similar in Table 1. Some data have moved out, it is not entirely clear what they relate to?

Response:

Sorry for the confusing. Table 1 shows the effects of zinc source and added zinc level on immunoglobulin and IL-6 secretion levels in 74-, 88-, and 102-day-old Xueshan Chickens under heat stress. Table 2 shows the effect of zinc source and added zinc level on duodenum morphology of 74-, 88-, and 102-day-old Xueshan chickens under heat stress.

  1. Have you taken into account the level of zinc in the feed?

Response:

Yes, we have taken into account the zinc level in the feed. The basal diet without Zn supplementation was used as the negative control group, and 50 mg/kg inorganic ZnS to meet the Zn nutritional requirements of 61–102-day-old Xueshan chickens was used as the positive control group.

  1. Regarding the discussion, the mechanism of action of your supplements and how they work is not explained.

Response:

Sorry for the confusing. While this study investigated the effect of zinc source and levels on the intestinal immunity of heat-stressed Xueshan chickens, we didn’t aim at a mechanistical understanding about the effect of zinc. We have already described why zinc play important roles for regulating immunity including citing multiple reference papers, thus we didn’t try to discuss further about the underlying mechanisms.

  1. Conclusions need to be drawn so that they are understandable to the public.

Response:

Thanks for your advice. We have drawn our conclusions at the end of the manuscript, as below:

In summary, this study investigated the effects of zinc source and level on the intestinal immunity of Xueshan chickens under heat stress. Our results indicate that adding specific sources and levels of zinc to the feed can improve intestinal immune function under heat stress conditions. Specifically, Zn-Prot M is more effective than ZnS, with an optimal supplementation level of 60 mg/kg.

  1. What is the reason that you have chosen this breed as the object of study of chickens.

Response:

China is the second largest chicken producer in the world, and Xueshan chickens is a new breed in China which showed increasing commercial importance due the distinct flavor of their meat; However, common broiler feeding standards may not meet the actual needs of Xueshan chickens, and there is an urgent need to improve the feeding standards for this specific breed. For these reasons, were chosen to study Xueshan chicken in this study.

  1. It is advisable to also indicate the productivity of poultry in the article, because in my opinion, all studies are indirectly or directly aimed at improving or increasing productivity.

Response:

Thanks for your suggestion. We agree that growth performance is important. However, this study focused on the intestinal immunity after heat stress, and the growth performance data are not fully documented since. We and collaborators are performing additional studies on the effect of zinc to Xueshan chickens, and we will pay more attention about the growth performance.

  1. The title of the article should be done differently for you more narrowly.

Response:

Thanks for your advice. We believe that the title of the current article has been able to concisely and accurately cover the research content.

  1. The list of references should be drawn up according to the requirements of the journal.

Response:

Thanks for your advice. We have revised the references according to the requirements of the journal.

Reviewer 3 Report

GENERAL COMMENT:

I consider this work is within the scope of “Animals”. It contains information useful in a field in which available information on the breed used is scarce and of special interest for searching alternatives to protects chicken against heat stress. Overall, it is well organised. I indicate below only minor points to be improved in the manuscript.

TITLE:

It is OK.

SIMPLE SUMMARY:

It is OK.

ABSTRACT:

It is OK.

KEYWORDS:

I suggest adding a keyword: “autochthonous breed”, or “local breed”.

INTRODUCTION:

Overall, this section is OK. However, some improvement is needed.

Please include some background on the rationale or interest to carry out this research on the Xueshan chicken breed.

Moreover, please add a bibliographical citation of Xueshan chicken breed. It is not an internationally known breed and when one perform a search on the Internet, very little information about the breed appears. This bibliographical citation or web link on the breed will permit the potential readers to learn about this breed.

MATERIALS AND METHODS:

Overall, this section is OK. However, several improvements are needed.

Line 94: Please include a bibliographical citation of "Laboratory Animal Management Regulations".

Line 98: Correct English writing: replace “lasted” with “lasting”.

Line 112: Please add a bibliographical citation for: “requirements of Chinese yellow-feathered broilers (2020)”

Lines 124 and 129: Type “ad libitum” in italics.

Line 126: Please revise whether “replicates of 16 chickens” must be: “replicates of 8 chickens

Lines 130-131: Please add bibliographical reference or web link for “Xueshan 130 chickens Feeding and Management Manual”.

Line 141: Indicate how many hours lasted the fasting period.

What about chicken sex? It is necessary to indicate whether the trial was carried out with a single sex or mixed sexes, and in this last case it is necessary to indicate whether sex effect was analysed or not.

RESULTS SECTION:

Overall, this section is OK.

DISCUSSION SECTION:

Overall, this section is OK.

CONCLUSIONS:

This section is OK.

REFERENCES SECTION:

In general terms, this section is well organised and adjusted to the style of the journal for references. However, I recommend reviewing it for removing typos and correct several flaws. For example:

Latin names of the organisms must be typed in italics. For example, Line 332-333: Salmonella enteritidis, and more.

TABLES:

Tables are OK.

Author Response

Response to reviewer’s comments:

Keywords:

I suggest adding a keyword: “autochthonous breed”, or “local breed”.

Response:

Thanks for your advice. We have added the keyword “local breed”.

Introduction:

  1. Please include some background on the rationale or interest to carry out this research on the Xueshan chicken breed.Moreover, please add a bibliographical citation of Xueshan chicken breed. It is not an internationally known breed and when one perform a search on the Internet, very little information about the breed appears. This bibliographical citation or web link on the breed will permit the potential readers to learn about this breed.

Response:

Thanks for your insightful comment. We have provided more descriptions about Xueshan chickens to the introduction section, and cited one paper which can be referred by the readers. Below is the newly added sentence:

Xueshan chickens is a new variety in China developed by crossbreeding high-quality local Tibetan chickens and Chahua chickens [1]. Its commercial importance is increasing due to its unique flavor and texture of the meat.

  1. Qin, S. Xueshan chickens Breeding System. Rural Prosperity. 2019, 9, 24-25.

Materials and Methods:

Line 94: Please include a bibliographical citation of "Laboratory Animal Management Regulations".

Response:

We have modified this sentence as suggested.

Line 98: Correct English writing: replace “lasted” with “lasting”.

Response:

We have replaced “lasted” with “lasting” as suggested.

Line 112: Please add a bibliographical citation for: “requirements of Chinese yellow-feathered broilers (2020)”

Response:

We have cited a relevant paper in our manuscript, as below:

The nutritional requirements for broiler chicks recommended in “Nutrient Requirements of Yellow Chickens (NY/T 3645–2020)” formulated by the National Technical Committee 274 on Animal Husbandry of Standardization Administration of China [2].

  1. National Technical Committee 274 on Animal Husbandry of Standardization Administration of China. Nutrient requirements of yellow chickens, NY/T3645–2020.; China Agriculture Press: Beijing, China, 2020.

Lines 124 and 129: Type “ad libitum” in italics.

Response:

We have italicized “ad libitum” as suggested.

Line 126: Please revise whether “replicates of 16 chickens” must be: “replicates of 8 chickens”

Response:

Sorry for this typo. It should be “replicates of 8 chickens”, and we have corrected it.

Lines 130-131: Please add bibliographical reference or web link for “Xueshan chickens Feeding and Management Manual”.

Response:

We decided to delete this sentence, since the manual is only in Chinese.

Line 141: Indicate how many hours lasted the fasting period. What about chicken sex? It is necessary to indicate whether the trial was carried out with a single sex or mixed sexes, and in this last case it is necessary to indicate whether sex effect was analysed or not.

Response:

Thanks for your advice. At the ages of 74, 88, and 102 days during the experiment, after a 12 h overnight fasting (20:00 pm to 8:00 am the next day) without restricting water intake. Xueshan chickens are new varieties selected from Tibetan chickens and camellia chickens by a variety of crosses. This study only used male chickens for experiment, since male Xueshan chicken has more meat flavor and greater commercial importance.

References:

In general terms, this section is well organised and adjusted to the style of the journal for references. However, I recommend reviewing it for removing typos and correct several flaws. For example: Latin names of the organisms must be typed in italics. For example, Line 332-333: Salmonella enteritidis, and more.

Response:

We have double-checked the references to remove any typos and flaws.
